# ABEL: Sample Efficient Online Reinforcement Learning for Neural Theorem Proving

**Fabian Gloeckle**
FAIR at Meta
École des Ponts Paris

**Jannis Limperg**
LMU Munich

**Gabriel Synnaeve**
FAIR at Meta

**Amaury Hayat**
École des Ponts Paris

## Abstract

We propose a scalable and efficient reinforcement learning framework as a strong baseline for theorem proving with limited data. This baseline reaches performances comparable to the current state-of-the-art in theorem proving with 59.8 % of problems solved on MiniF2F-*valid* cumulatively and the current state of the art of 7/640 solved problems on PutnamBench, while only training on a few hundred examples in the reinforcement learning set. This a first step toward an efficient and easily reproducible combination of autoformalization, synthetic data generation and reinforcement learning, which could unlock significant advancements in neural theorem proving.

## 1  Introduction

Mathematical reasoning constitutes a major challenge for deep-learning models, and now a very active research area [Williamson, 2024]. Formal languages such as Isabelle [Paulson, 1994], Coq [Barras et al., 1997], and Lean [de Moura et al., 2015, Moura and Ullrich, 2021] have been developed to enable automatic computer verification of proofs and can now serve as grounding to prevent language model hallucinations. Several approaches relying on LLMs and formal proof search environments have recently been proposed (App. A), but were limited by the scarcity of formal training data (around 100k lemmas in Lean's central theorem library *Mathlib* [Mathlib Community, 2020]) and the data inefficiency of machine learning methods. For this reason, most of the works on neural theorem proving have focused on obtaining more data, either from autoformalization or synthetic data generation [Xin et al., 2024a].

In this paper, we present ABEL, a scalable and compute efficient online reinforcement learning framework for theorem proving. This serves as a strong baseline of what is achievable with very limited data and as a first step toward a combination of online reinforcement learning and autoformalization.

We show that training on only several hundred maths exercises, one can reach performances comparable to the current state-of-the-art in theorem proving. We use MiniF2F [Zheng et al., 2021], a well-established benchmark in the field, as our primary evaluation set. Our model outperforms the cumulative performances of [Lample et al., 2022] with 13 times less compute and no synthetic data. It also reaches a new state-of-the-art on PutnamBench [Tsoukalas et al., 2024b], a dataset of formidably challenging olympiad-like problems from the Putnam competition in North America, by solving 7 problems overall and discovering one formalization error, 4 more than the previous best [Tsoukalas et al., 2024a].

This suggests that combining our framework with autoformalization and synthetic data generation could achieve much higher levels of performance and unlock significant advancements in neural theorem proving.

38th Conference on Neural Information Processing Systems (NeurIPS 2024).

## 2 Method

When approaching theorem proving as a problem of reinforcement learning, we must take care to cast the problem in a way that allows for efficient exploration and recombination of learned features. In this work, we follow the formulation of *hypertree proof search* (HTPS) [Lample et al., 2022] and opt for a two-fold tree structure for proof searches. An individual complete Lean proof is naturally represented as a tree, where nodes correspond to proof goals and tactics like `induction` transform a proof goal into a set of children, or subgoals, that *jointly* suffice to be shown instead (e.g. induction base and step). Proof search, on the other hand, involves applying several tactics at each node, proof success of *any* of which results in proof success for the parent node. Such a tree can be represented as an alternating tree of AND-joined sets of subgoals and OR-joined sets of tactic attempts, making the overall tree an AND/OR-tree or, equivalently, a "proof hypertree"[1].

Our reinforcement learning system for formal theorem proving in Lean 4 [Moura and Ullrich, 2021] consists of three components: a programming interface based on Aesop [Limperg and From, 2023] to organize proof searches in Lean 4, the HTPS proof search procedure inspired by the AlphaZero expert iteration algorithm [Anthony et al., 2017, Silver et al., 2018] and an online retraining mechanism. We will explain each component target the following paragraphs.

**AesopRepl: a proof search interface for Lean 4**   Our Read-Eval-Print Loop (REPL), AesopRepl, makes the infrastructure of Lean's proof search automation tactic Aesop [Limperg and From, 2023] available as a command line interface. Aesop manages an AND/OR-proof tree and can receive tactic suggestions generated by a language model via the interface. It executes the tactics in their respective contexts and updates the tree, checking which goals have been resolved. Once the root node of the proof tree (i.e., the initial goal) is proved, the REPL passes the resulting proof to Lean's kernel. This guards against bugs in the applied tactics. The main **benefit** of using Aesop's proof tree implementation is that it allows goals with shared metavariables to be processed semi-independently, improving on GPT-$f$ [Polu and Sutskever, 2020] and LeanDojo [Yang et al., 2023] and matching HTPS [Lample et al., 2022]. See App. J for details.

**Proof search with HTPS**   Our proof search procedure uses the AlphaZero algorithm [Silver et al., 2018] with the adaptations for hypertrees proposed by [Lample et al., 2022]. A proof search consists of several rounds, each of which has three steps: *node selection*, *expansion* and *value propagation*. The proof search is terminated when either a proof has been found during a round or a given node expansion budget is exceeded. In the selection phase, the tree is traversed down until a solving set of leaf nodes is found, i.e. a set of nodes which, if proven, would finish the overall proof. This means that at AND-nodes, all children are selected while at OR-nodes, we select a tactic based on the established predictor upper confidence bound for trees (PUCT) policy [Kocsis and Szepesvári, 2006, Rosin, 2011, Silver et al., 2017] given below. In the expansion phase, all selected nodes are expanded simultaneously, i.e. a tactic model suggests tactics and a critic model evaluates the value (likelihood of proof success) of the given proof goal. We execute the tactics eagerly in Lean in order to return from successful proof searches as early as possible. The transition dynamics $T(s, a) = s'$ of Lean, where $s$ is a state/goal and $a$ an action/tactic, i.e. the effect of a tactic execution, are sometimes hard to model (tactics like `simp` involve a search procedure themselves), so we follow HTPS in using a state value critic $c(s')$ as opposed to a state-action critic $Q(s, a)$ in AlphaZero [Silver et al., 2018]. These *critic* values of expanded nodes are then propagated up the tree: each edge $(s, a)$ maintains a visit count $N(s, a)$ and a cumulative action value $W(s, a)$. Each node $s$ in the selected proof tree is assigned an update value $v(s)$ which is defined as follows:

$$v(s) = \begin{cases} 1 & \text{if } s \text{ is proved,} \\ c(s) & \text{if } s \text{ was just expanded,} \\ \gamma \prod_{c \text{ child of } s} v(c) & \text{otherwise,} \end{cases}$$

where $\gamma \in [0, 1]$ is a depth penalty factor and $c(s)$ denotes the evaluation provided by the critic model. The cumulative action value $W(s, a)$ of each edge $(s, a, s')$ in the selected tree is then increased by the update value $v(s')$ of its target node, and its visit count by 1. With these quantities, we can define

---

[1]when considering the set of children of a tactic's child AND-node the collective target of a hyperedge

the PUCT selection policy:

$$\mathrm{PUCT}(s) = \operatorname*{argmax}_a \hat{Q}(s,a) + c_{\mathrm{puct}} \cdot \pi(a \,|\, s) \frac{\sqrt{\sum_{a'} N(s,a')}}{1 + N(s,a)},$$

where

$$\hat{Q}(s,a) = \begin{cases} \frac{W(s,a)}{N(s,a)} & \text{if } N(s,a) > 0, \\ c(s) & \text{otherwise} \end{cases}$$

is the empirical action value of the edge $(s,a)$ and $c_{\mathrm{puct}}$ is an exploration coefficient and $\pi(a \,|\, s)$ is the prior of action $a$ under our tactic model $\pi$, for which we take the product of token probabilities of each tactic sequence, renormalized to sum to one at any node. Note that in our choice of $\hat{Q}(s,a) = c(s)$ for unvisited nodes we depart from Lample et al. [2022] (who choose $\hat{Q}(s,a) = \frac{1}{2}$), leveraging the intuition that $c(s)$ is trained to match the expected proof success rate when performing tree search using $\pi$, and that $a$ is a sample from $\pi$.

Following Lample et al. [2022], instead of fixing proof search hyperparameters upfront, we sample them on a per proof attempt basis (App H).

**Online reinforcement learning**   When a proof search is terminated (because a proof has been found or the search budget is exhausted), we extract training samples for the policy and critic models for retraining. We train the **policy model** with supervised training using standard causal language modelling loss on all tactic samples that are part of a proof of their parent node – regardless of whether they were part of a proof of the root node (*All Solved* setting of Lample et al. [2022]). This makes the tactic reinforcement learning loop a form of iterated rejection sampling.

The **critic model** is trained with supervised training using a binary cross-entropy loss of Bernoulli variables for the classification task of provability. As the supervision target, we use $\mathrm{Bernoulli}(V(s))$ with $V(s)$ given by

$$V(s) = \frac{W(s,a^*)}{N(s,a^*)},$$

for a node $s$ and $a^* = \mathrm{PUCT}(s)$. Unlike Lample et al. [2022], we do not threshold the visit counts $\sum_a N(s,a)$ of a node to be selected.

Our **distributed reinforcement learning** setup comprises worker and trainer GPUs, prover threads controlling Lean AesopRepl processes as well as a centralized replay buffer and a task dispenser (App. D). We select proof tasks with **prioritized sampling** to favor exploration: if $n_i$ is the number of successful proof searches for problem $i$ so far, we use a weight proportional to $(n_i + 1)^{-\alpha}$ for sampling problem $i$ as the next task, where $\alpha \geq 0$ is an upsampling coefficient for hard tasks[2]. We **postprocess tactics** to induce a distribution shift from style aimed at efficient *proof presentation* as pursued in Lean's Mathlib [Mathlib Community, 2020] to effective *proof search* (App. E).

## 3   Results

We conduct experiments on challenging theorem proving datasets, showcasing the efficiency of our theorem proving system. We use Llama 3.1 base 8B [Dubey et al., 2024] as a starting point and finetune on proof step data from Mathlib [Mathlib Community, 2020] extracted with LeanDojo [Yang et al., 2023] (App. F for details). We then apply the RL training procedure described in Section 2. Our results are summarized in Table 1.

Our models' cumulative performance, while below other approaches relying on large sets of data, outperforms HTPS [Lample et al., 2022] despite using a much smaller compute budget and no synthetic data. Our models also show high pass@1 performances on the test set, which are only surpassed by DeepSeek-Prover, and achieve a new state-of-the-art on PutnamBench. This is detailed below.

---

[2] $\alpha = 0$ reduces to uniform sampling, $\alpha = +\infty$ to uniform sampling among least often solved problems

Table 1: **Performance of different models on MiniF2F-*valid* and MiniF2F-*test*.** We compare representative methods without reinforcement learning with recent expert iteration and online reinforcement learning approaches. Numbers on MiniF2F-*test* are with the indicated evaluation budget given as $n_{\text{attempts}} \times n_{\text{expansions}} \times n_{\text{tactics}}$. For MiniF2F-*valid*, we report *cumulative* solve rates over the course of the run where it is included in the reinforcement learning set. We convert train times to A100-days with a factor 3 for H100 to A100 performance [Databricks, 2023]. For additional discussion and details, see App. K.

| Model | Type | test *eval.* | Budget | valid *cumul.* | Train time (A100-days) | RL set size |
|---|---|---|---|---|---|---|
| Llemma-7b | pretr. | 26.2 | $1 \times 600\,\text{s} \times 32$ | – | (1000) | – |
| ReProver | SFT | 26.5 | $1 \times 600\,\text{s} \times 64$ | – | 5 | – |
| Lean automation | alg. | 27.5 | $1$ | 26.6 $_{\text{eval}}$ | – | – |
| DeepSeek-Prover | exp.it. | 48.8 | $8192$ | 60.2 | ? | 8M |
| DeepSeek-Prover-1.5 | exp.it.+RL | 55.0 | $1 \times 3200$ | – | ? | 8M |
| GPT-$f$ | exp.it. | 36.6 | $64 \times 512 \times 8$ | 47.3 $_{\text{eval.}}$ | 2000 | 327 + *synth.* |
| HTPS | RL | 41.0 | $64 \times 5000 \times 26\,\text{avg.}$ | 58.6 | 1360 | 244 + *synth.* |
| *ABEL (ours)* | RL | 41.3 | $1 \times 128 \times 64$ | 59.8 | 96 | 244 |

**Outperforming HTPS with 13x less compute**   Training with 256 GPUs on MiniF2F-*valid*, we reach a cumulative solve rate of 59.8% after 3 hours of training time, outperforming HTPS's 58.6% trained with an additional synthetic supervised dataset of Lean problems and 54.9% without. These results required HTPS 30240 A100-hours of compute, while we need 768 H100-hours $\approx$ 2304 A-100-hours [Databricks, 2023], approximately 13 times less. Halving the number of GPUs to 128, we reach a cumulative solve rate of 57.4% after 5 hours, still competitive with HTPS.

**A new state of the art on PutnamBench**   When run on a 1:1:1 mix of problems from MiniF2F-*valid*, MiniF2F-*test* and PutnamBench [Tsoukalas et al., 2024b], we solve 7/640 problems in the setting with provided solutions after only attempting each problem between 6 and 16 times (and with no improvements thereafter with up to 590 attempts). This improves over the previous state of the art of 4/640 solved problems [Tsoukalas et al., 2024a] held by InternLM2-StepProver [Wu et al., 2024] and obtained with 4096 attempts per problem. In a different run, our system solved another problem, bringing the total to 8/640. See App. B for the proofs, a **qualitative evaluation** and the cumulative solve rates on the respective datasets.

**Importance of online training**   We evaluate the performance of pure sampling, expert iteration [Anthony et al., 2017, Polu et al., 2022], and online reinforcement learning by comparing runs with model updates at different frequencies. Our findings indicate that online reinforcement learning is crucial (App. C).

**Stability and exploration**   Reinforcement learning on MiniF2F-*valid* – which comprises 244 problems only – presents challenges regarding stability and distributional collapse. In our online training loop, this becomes exacerbated by the fact that at least 26.6% of problems in MiniF2F can be solved by a single or a short sequence of automation tactics (among `aesop`, `ring`, `linarith`, `nlinarith`). Such proofs will be found first and could quickly flood the trainers low-quality data that lacks diversity. We remediate this issue in several ways: by setting a *burn-in* of 8000 training samples to discard before training, by using supervised data for 10% of training samples, by using hard negative sampling (Sect. 2) by using a large number of tactics per node and decoding parameters that favor diversity as well as carefully tuning key hyperparameters (App. I). As shown in Fig. 1, the policy continues to explore and optimizes solved problems to shorter proofs, but we still see a slow decline in tactic diversity when problem solve rates eventually plateau.

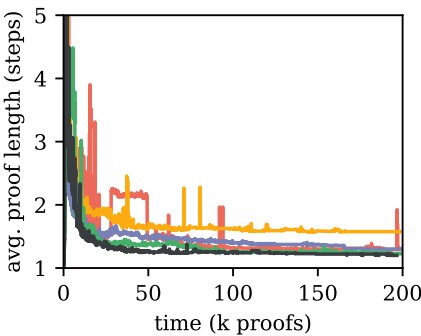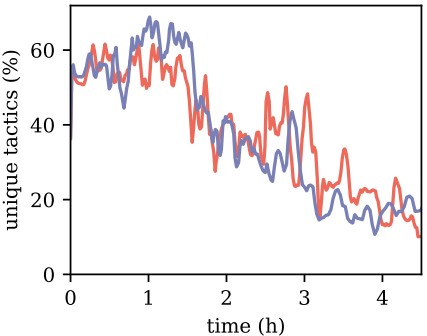

Figure 1: **Evolution of average proof length (left) and tactic diversity (right) over the course of training.** Running average of the number of steps of the most recent proof per problem, indexed by the total number of proofs found so far. Over the course of each of the five depicted runs, models "golf" proofs to more compressed versions. At the same time, the policy loses diversity: the proportion of unique tactics among the total number of tactics sampled at each expansion declines (two runs).

## 4   Conclusion

We presented an online reinforcement learning method for Lean that reaches near state-of-the-art performance on theorem proving while relying only on few hundred problems in its reinforcement learning problem set. It is orders of magnitude more sample efficient than methods of comparable performance that require extensive synthetic datasets with up to millions of additional data points [Xin et al., 2024a], and around 13 times more compute efficient than the current state-of-the-art RL method [Lample et al., 2022].

Online reinforcement learning can be tricky to stabilize but carefully tuned, complements and empowers other recent techniques such as synthetic data from large-scale autoformalization [Xin et al., 2024a], bootstrapping chains-of-thought [Lin et al., 2024] or hybrid local-global tree search [Xin et al., 2024b]. In future work, we believe the setup should be adapted to *condition* the policy on the node's ancestors, i.e. on the partial proof to be completed. When deploying automated provers for *theory autoformalization*, additional conditioning on a human-written or language model generated informal proof is additionally required, for instance in a setting similar to Draft-Sketch-Prove [Jiang et al., 2023].

Harnessing *reinforcement learning as a test-time inference technique* could allow tackling challenging problems such as the International Mathematics Olympiad [DeepMind, 2024] or autoformalizing mathematical theories, where sample efficiency is the primary concern.

## Acknowledgments

We thank, in no particular order, Aram Markosyan, Jonas Gehring, Vegard Mella, Quentin Carbonneaux, Badr Youbi Idrissi, Mathurin Videau, Robin San Roman, Gwenaëlle Léon, Kaiyu Yang, Johan Commelin, Julia Kempe, Yunzhen Feng, Olivier Teytaud, Guillaume Lample, Timothée Lacroix and all FAIR PhD students, CodeGen team members and reasoning team members for helpful and enriching discussions and technical support.

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

## Supplementary Material

## A    Related works

Mathematics is often seen as the culmination of reasoning tasks because of its intricacy and level of abstraction. While deep-learning approaches enabled mathematical discoveries in the last few years [Davies et al., 2021, Douglas, 2022, Blundell et al., 2022, Alfarano et al., 2024], it was used merely as a tool where the reasoning is left to the human. In the recent years many neural approaches have been proposed for neural theorem proving. These include, among others, fine-tuning a language model used to guide a step-by-step proof search [Polu and Sutskever, 2020, Polu et al., 2022, Azerbayev et al., 2024, Wu et al., 2024] or to generate whole proofs [First et al., 2023]; autoformalization of natural language data to either bridge the link between language models and classical automated theorem proving tools [Jiang et al., 2023] or increase the quantity of available data for training a model [Xin et al., 2024a]; combination of fine-tuned model and retriever to suggest relevant context from a library [Yang et al., 2023]; reinforcement learning (RL) [Lample et al., 2022] inspired from AlphaZero [Silver et al., 2018].

## B    Sample proofs found on PutnamBench

```
1  theorem putnam_1988_b1
2  : ∀ a ≥ 2, ∀ b ≥ 2, ∃ x y z : ℤ, x > 0 ∧ y > 0 ∧ z > 0 ∧
3                              a * b = x * y + x * z + y * z + 1 := by
4  aesop
5  exact ⟨a - 1, by linarith, b - 1, by linarith, 1, by norm_num, by ring⟩
6
7  theorem putnam_1986_a1
8  (S : Set ℝ)
9  (hS : S = {x : ℝ | x ^ 4 + 36 ≤ 13 * x ^ 2})
10 (f : ℝ → ℝ)
11 (hf : f = fun x ↦ x ^ 3 - 3 * x)
12 : (∀ x ∈ S, f x ≤ ((18) : ℝ )) ∧ (∃ x ∈ S, f x = ((18) : ℝ )) := by
13 norm_num <;> aesop
14 · cases le_total 3 x <;> nlinarith
15 · exact ⟨3, by norm_num⟩
16
17 theorem putnam_2001_a1
18 (S : Type*)
19 [Mul S]
20 (hS : ∀ a b : S, (a * b) * a = b)
21 : ∀ a b : S, a * (b * a) = b := by
22 intro a b
23 convert hS _ _ using 2 <;> simp <;> ring
24 rw [hS ]
25
26 theorem putnam_1988_b2
27 : (∀ x y : ℝ, (y ≥ 0 ∧ y * (y + 1) ≤ (x + 1) ^ 2) → (y * (y - 1) ≤ x ^ 2))
28                              ↔ ((True) : Prop ) := by
29 aesop
30 cases le_total x y <;> cases le_total y x <;>
31   simp_all <;> nlinarith
```

**Qualitatively evaluating** the last proofs found by the model, we observe: One statement had a formalization mistake, rendering the problem trivial. Other problems were easy compared to other Putnam problems because the solution was provided. In all cases, our model relies heavily on Lean's automation tools `simp`, `ring`, `omega`, `nlinarith`, `linarith`, but provides arguments and data without which the automation would not succeed (a classic pattern in idiomatic Lean proofs).

Cumulative solve rates on the three datasets used in the runs are reported in Table 2.

Table 2: **Results of runs with mixed reinforcement learning set.** Cumulative solve rates on the respective datasets after 7 hours runtime.

| Dataset | PutnamBench 1 | PutnamBench 2 |
|---|---|---|
| MiniF2F-*valid* | 55.3 | 58.6 |
| MiniF2F-*curriculum* | 27.7 | 32.3 |
| PutnamBench | 7/640 | 6/640 |

## C  Importance of online training

We compare runs with online model updates every 50 training steps, to a setting resembling batch reinforcement learning or expert iteration (as used in [Polu et al., 2022, Xin et al., 2024a]) with updates every 500 steps and a pure supervised model solving benchmark problems without any retraining. Figure 2 shows the cumulative proof rates on MiniF2F-*valid* for the respective runs each using 56 workers and 8 trainers (if applicable). The online reinforcement run clearly outperforms the others. Note that in our final runs, we update the weights even more frequently, namely every 10 steps.

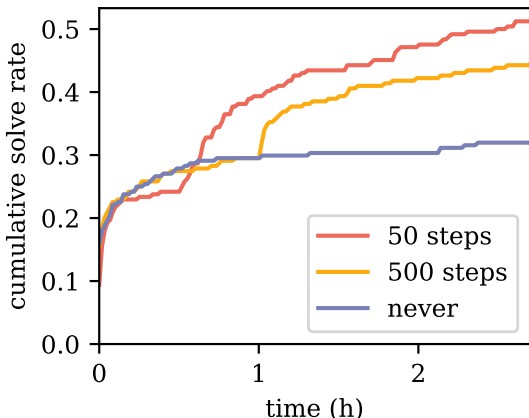

Figure 2: **Importance of online training.**

## D  Distributed reinforcement learning

Our distributed setup comprises *worker* GPUs for language model inference, *trainer* GPUs for language model training and *prover* CPU threads orchestrating proof searches and controlling Lean AesopRepl processes. In our experiments, we assign 10 prover threads to each worker GPU, enabling stable generation inference workloads in our GPU-bound setup. A central service distributes proof tasks according to the hard negative sampling policy described in Sect. 2. Workers send their tactic and critic training samples to a centralized replay buffer which distributes them to the trainers, prioritizing samples seen fewer times and from more recent model generations. Unlike [Lample et al., 2022], we do not parallelize within individual tree searches but only at the level of proof attempts (i.e. we apply node selection and expansion sequentially rather than asynchronously).

## E  Induced distribution shift via tactic post-processing

The optimal policy for proof search naturally differs from the compressed proofs found in supervised finetuning data. For instance, proofs in Lean's `Mathlib` [Mathlib Community, 2020] often need to specify the full set of simplification lemmas used in the `simp` tactic without relying on standard configurations which are prone to change over time, possibly breaking existing proofs. Since in this style, proofs are tedious to find, Mathlib authors use automatic tooling to convert proofs using `simp`'s standard configuration into maintainable, fully specified `simp only` proofs. Likewise, Mathlib authors "golf" sequences of rewriting rules into a single `rewrite` tactic with multiple arguments.

To remediate this distribution discrepancy between *proof search* and *proof presentation*, we add post-processing to the tactics generated by the policy model. This facilitates distribution shift toward a better *proof search policy*: we turn `simp only` into `simp`, add a single-rule `rewrite` with the first rule for each multi-rule `rewrite` and deduplicate `simp` lemmas to break model repetitions.

## F  Finetuning hyperparameters

For supervised finetuning on the proof step dataset extracted from Mathlib (see App. L), we train with a peak learning rate of $10^{-4}$ and a total batch size of 16k tokens using a block-causal attention mask that does not split sequences. We use the Adam optimizer [Kingma and Ba, 2015] with decoupled weight decay [Loshchilov and Hutter, 2019] with a peak learning rate of $10^{-4}$, $\beta_1 = 0.9$, $\beta_2 = 0.95$ and $L_2$ weight decay coefficient 0.1. We schedule the learning rate with cosine decay [Loshchilov and Hutter, 2017] over 5000 steps to a final learning rate of $10^{-6}$.

## G  Training format and inference techniques

Our supervised training on Mathlib data uses a **format** containing theorem name, Mathlib file path, the theorem statement, the current proof goal, a placeholder for the critic, the tactic applied in the proof and the outcome of the tactic application. In other words, we condition the tactic policy on the mentioned fileds and train with a tactic outcome prediction co-training task [Gloeckle et al., 2023]. Example for the format:

```
1   NAME: MvPolynomial.prod_X_add_C_coeff
2   FILE: Mathlib/RingTheory/Polynomial/Vieta.lean
3   STATEMENT: theorem MvPolynomial.prod_X_add_C_coeff (k : ℕ) (h : k ≤ card σ) :
4       (∏ i : σ, (Polynomial.X + Polynomial.C (MvPolynomial.X i)) : Polynomial _).coeff k =
5       MvPolynomial.esymm σ R (card σ - k)
6   STATE: R : Type u_1
7   σ : Type u_2
8   inst†¹ : CommSemiring R
9   inst† : Fintype σ
10  k : ℕ
11  s : Multiset (MvPolynomial σ R) := Multiset.map (fun i => X i) univ.val
12  h : k ≤ Multiset.card s
13  this : Fintype.card σ = Multiset.card s
14  ⊢ (Multiset.map (fun i => Polynomial.X + Polynomial.C (X i)) univ.val).prod.coeff k =
15      (Multiset.map X univ.val).esymm (Multiset.card s - k)
16  PROVABLE: 1
17  TACTIC: convert Multiset.prod_X_add_C_coeff s h
18  RESULT: case h.e'_2.h.e'_3.h.e'_3
19  R : Type u_1
20  σ : Type u_2
21  inst†¹ : CommSemiring R
22  inst† : Fintype σ
23  k : ℕ
24  s : Multiset (MvPolynomial σ R) := Multiset.map (fun i => X i) univ.val
25  h : k ≤ Multiset.card s
26  this : Fintype.card σ = Multiset.card s
27  ⊢ Multiset.map (fun i => Polynomial.X + Polynomial.C (X i)) univ.val =
28      Multiset.map (fun r => Polynomial.X + Polynomial.C r) s
```

For **inference**, we use the same template up to "provable:" and apply joint critic and policy inference from a shared model as follows: at the position after "provable:", we restrict the output vocabulary to two designated tokens and compute the critic value $c(s)$ as the logit softmax

$$c(s) = \text{sigmoid}(x(s)_1 - x(s)_0),$$

where $x(s)$ denotes the transformer's output logits at the respective position, and "0" and "1" denote the respective token indices. We then append a "1" at the critic's position and continue with autoregressive sampling to retrieve the policy's tactic prediction. By sharing the model, we aim to reap knowledge transfer between the critic and the policy task, following standard practice in

reinforcement learning [Silver et al., 2018]. With our joint inference mechanism, we can, moreover, leverage existing autoregressive language model generation pipelines at the expense of a single additional token.

As an additional **inference speed optimization**, we compute the key-value cache of the (typically long) prompt only once during the "pre-fill phase" before copying it as many times as we would like to decode independent tactic suggestions. Using this method, we observed a 2-3x inference speedup. We stop decoding at a special stop token between the "tactic:" and the "result:" field not shown above.

## H   Proof search hyperparameters

Following Lample et al. [2022], we sample the decoding and proof search hyperparameters before each proof search as follows:

- **temperature** for nucleus sampling [Holtzman et al., 2020] with probability mass 0.9: uniform on $[0.8, 2.4]$ (HTPS uses $[0.8, 2.0]$ and standard sampling),
- **number of expansions** (the maximum number of goals for which to decode tactics during a proof search): log-uniform on $[404, 4047] \approx [128\sqrt{10}, 1280\sqrt{10}]$ (HTPS uses $[1000, 10000]$),
- **depth penalty** $\gamma$: uniform from $\{0.8, 0.9, 0.95, 1.0\}$,
- **PUCT exploration coefficient** $c_{\text{puct}}$: log-uniform on $[0.01, 100]$.

Unlike HTPS, we always sample 64 tactics per node expansion with the understanding that tactic decoding is fast compared to the encoding (kv-cache pre-fill) of the proof state prompt.

## I   Reinforcement learning configuration

Table 3: **Hyperparameters for the runs reported in Sect. 3.**

| Parameter | 128 GPU run | 256 GPU run | PutnamBench 1 | PutnamBench 2 |
|---|---|---|---|---|
| workers | 120 | 248 | 248 | 248 |
| buffer max. sends | 3 | 1 | 1 | 2 |
| nucleus sampling mass | 0.9 | 1.0 | 1.0 | 1.0 |
| negative sampling coefficient $\alpha$ | 0.3 | 0.5 | 0.5 | 0.5 |

All runs use 8 trainer GPUs, a learning rate of $4 \cdot 10^{-5}$, batches of 4 sequences of 4096 tokens per GPU, a maximum buffer size of $10^6$, a rehearsal rate of $0.1$, weight updates every 10 training steps and 8000 *burn-in* samples discarded at the beginning of the run. All other hyperparameters are detailed in Table 3.

The **replay buffer** uses the following selection and eviction policy: it sorts samples by the smallest number of times they have already been sent to the trainers, then by the most recent model version they come from, and randomly for ties, and sends them by increasing priority. Samples are evicted from the buffer if they have been sent a certain maximum number of times or if the buffer has reached its maximum capacity, the sample has been sent at least once and it is lowest in the buffer's priority ordering.

## J   Aesop and shared metavariables

Aesop's proof tree implementation is very similar to that HTPS [Lample et al., 2022], with one exception concerning the handling of shared *metavariables*. These are Lean constructs representing objects that will be filled in during the proof. For example, a transitivity tactic might reduce the goal $\vdash 1 < 3$ to two goals $\vdash 1 < ?x$ and $\vdash ?x < 3$, where the metavariable $?x$ stands for a number to be determined later. Another tactic application could then solve the first goal for $?x = 2$, at which point $?x$ is replaced with 2 in the second goal as well.

The example demonstrates the central issue with metavariables: If the proof of the first goal had assigned $?x = 4$, the second goal would have become unprovable. In other words, the possible proofs of the second goal depend on which proof was chosen for the first goal (or vice versa); the goals are no longer independent. But independence of goals is a central assumption underlying the hypertree proof search model.

GPT-$f$ [Polu and Sutskever, 2020] and LeanDojo [Yang et al., 2023] address this issue by operating on tactic states, i.e. lists of goals. However, this reduces parallelism and leads to duplicate work since the same goal may have to be proved in multiple tactic states. HTPS refines the GPT-$f$ approach by using tactic states (with more than one goal) only when goals are, in fact, coupled by a shared metavariable. Aesop goes even further and treats coupled goals as independent, but when a metavariable is assigned, Aesop makes a copy of any goals affected by this assignment [Limperg and From, 2023, Sec. 4]. Hence, the doomed proof attempt that sets $?x = 4$ does not prevent a later successful proof attempt with $?x = 2$.

Compared with HTPS, Aesop's approach may **enable future optimisations** that transfer a tactic application from a goal to its copies if the tactic application is independent of any specific metavariable assignment. This would reduce the amount of effort spent during proof search on applying the same tactics to goals that differ only in their metavariable assignments. Unlike [Lample et al., 2022], we do not currently attempt to merge duplicate goals.

## K Notes on model comparison

In Table 1, we include representative methods without reinforcement learning (classical solvers, pretrained models with few-shot prompting, supervised finetuning) and recent reinforcement learning approaches.

**Lean automation** refers to a small search using the following Lean tactics: `aesop`, `ring`, `linarith` and `nlinarith`.

**ReProver** [Yang et al., 2023] is a language model coupled with a premise retrieval model which was trained on supervised data extracted from Mathlib.

**Llemma-7b** [Azerbayev et al., 2024] is a model based on CodeLlama [Rozière et al., 2023] that was trained on formal and informal mathematical data extacted from the web [Paster et al., 2023].

We do not include **InternLM2-StepProver** [Wu et al., 2024] which reaches 48.8% on MiniF2F-*test* with a budget of $1 \times 100 \times 32$ and 63.9% on MiniF2F-*valid*, but whose data collection raises concerns about data contamination. For instance, the Lean 3 MiniF2F dataset at `https://github.com/facebookresearch/miniF2F` contains solutions.

**DeepSeek-Prover** [Xin et al., 2024a] and **DeepSeek-Prover-1.5** [Xin et al., 2024b] do not report the total amount of compute used for developing their models. Based on the large numbers of problems in their reinforcement learning set, expensive evaluation settings of up to pass@65536 and the large number of models and methods reported in the publications , we expect it to be significantly larger than for the other models in the comparison. We also note that while all other models in the comparison are sized between 600M parameters (HTPS) and 8B parameters (ours), DeepSeek-Prover also uses sequence distillation from a 236B parameter model.

DeepSeek-Prover uses 8M autoformalized problems in its reinforcement learning set. DeepSeek-Prover-1.5 adds MiniF2F-*valid* (244 problems), ProofNet-valid [Azerbayev et al., 2023] (185 problems) and Lean Workbook [Ying et al., 2024] (57k problems, or 140k including "Workbook Plus").

**GPT-$f$** [Polu et al., 2022] uses MiniF2F-*curriculum* instead of MiniF2F-*valid* in the reinforcement learning set. The reported number on MiniF2F-*valid* is hence an evaluation number with the same budget as the number on MiniF2F-*test*, not a cumulative one.

**HTPS** [Lample et al., 2022] samples the number of tactics sampled at each node uniformly from $\{8, 16, 32, 48\}$, with an average of 26, which we report in the table. (Note however, that for pass@64, the maxmum may be more relevant than the average.)

Both GPT-$f$ and HTPS additionally use synthetically generated data. GPT-$f$ generates 5600 problems without proof as reinforcement learning tasks. HTPS generates problems including proofs, and uses the data as an additional supervised dataset instead.

## L   Dataset versions

We used the following commits of the respective datasets:

- Mathlib [Mathlib Community, 2020] for Lean4 (`https://github.com/leanprover-community/mathlib4/`): `29dcec074de168ac2bf835a77ef68bbe069194c5` (corresponds to LeanDojo [Yang et al., 2023] v10),
- MiniF2F [Zheng et al., 2021] for Lean4 [Yang, 2024] (`https://github.com/yangky11/miniF2F-lean4`): `6bcf0b4940fbf17a1ba83db4ed639fbcb26b1a27` (latest commit at the time of this paper),
- PutnamBench [Tsoukalas et al., 2024b] (`https://github.com/trishullab/PutnamBench`): `8aaccf8c40e0db1e69a3c808166d9d09b0109703` (latest commit at the time of this paper).

