# OpenReview forum: "ABEL: Sample Efficient Online Reinforcement Learning for Neural Theorem Proving"
_NeurIPS.cc/2024/Workshop/MATH-AI — MATH-AI 24_

### Official Review · Reviewer_mGak · 2024-10-02
**An Incremetnal Work of HTPS**

**Rating:** 6
**Confidence:** 4

**Review:**

The paper presents ABEL, a reinforcement learning framework designed for formal theorem proving that builds upon the previous work of HTPS. ABEL incorporates more advanced language models and is trained on the latest datasets, LeanDojo. Experimental results demonstrate that ABEL outperforms several state-of-the-art baselines, including DeepSeek Prover, GPT-f, and HTPS on the miniF2F and PutnamBench benchmarks.

The paper is well-written and easy to follow. While many core techniques (e.g., hypertree proof search, online training) are inherited from HTPS, ABEL integrates a better fine-tuned LLM and a relatively smaller real-world dataset within its reinforcement learning loop. This allows ABEL to surpass HTPS in cumulative performance, achieving these results with less computational overhead and without relying on synthetic data.

Minor: Reference 18 in the appendix has a missing author name (shown as `author?`).

---

### Official Review · Reviewer_9Zs6 · 2024-10-07
**Review of ABEL**

**Rating:** 5
**Confidence:** 3

**Review:**

Pros: The paper presents ABEL, an innovative reinforcement learning framework aimed at addressing the challenges of neural theorem proving with a strong emphasis on sample and computational efficiency. ABEL integrates hypertree proof search (HTPS) with online reinforcement learning techniques, drawing inspiration from the AlphaZero algorithm to deliver high performance on formal mathematics tasks. Key aspects of ABEL include a proof tree structure that enables efficient proof searches and an online retraining mechanism that continuously enhances model performance with new data. ABEL requires significantly fewer training examples than existing state-of-the-art models, making it a strong baseline for formal theorem proving tasks.

Cons: The experiments are conducted using only one model and a single setting, which limits the ability to demonstrate the generalizability and overall effectiveness of the proposed method

---

### Official Review · Reviewer_8Emm · 2024-10-08
**ABEL is a scalable and sample-efficient online reinforcement learning framework for theorem proving in Lean 4. It outperforms prior approaches while using significantly less compute and data. ABEL achieves a new state-of-the-art on the challenging PutnamBench dataset and opens promising directions for combining RL with auto-formalization and synthetic data generation.**

**Rating:** 7
**Confidence:** 4

**Review:**

The authors propose ABEL, a scalable and efficient online reinforcement learning framework for theorem proving in Lean 4. ABEL incorporates several key features, including efficient online RL using hypertree proof search (HTPS) and AlphaZero-style tree search, an interface layer (AesopRepl) between the RL system and Lean's Aesop automation system, optimizations for critic and policy training to improve sample efficiency and exploration, and a distributed RL setup for scalability. Experiments demonstrate that ABEL outperforms prior HTPS results with 13x less compute and no synthetic data. Furthermore, it achieves a new state-of-the-art on the challenging PutnamBench dataset.

Pros
- ABEL achieves competitive theorem-proving performance with orders of magnitude better sample efficiency than prior work.
- The paper introduces a scalable online RL setup tailored specifically for theorem proving, enabling rapid iterations and improvements.
- Several interesting technical innovations are presented around stabilizing online RL, such as hard negative sampling, burn-in, rehearsal, and tactic post-processing.
- The empirical results are strong, including establishing a new state-of-the-art on the difficult PutnamBench benchmark.
This work opens up promising new research directions around combining RL with auto-formalization and synthetic data generation to advance theorem-proving capabilities.

Cons
- The empirical evaluation is somewhat limited in scope, focusing primarily on the MiniF2F and PutnamBench datasets. Broader evaluation on additional datasets would strengthen the claims. This is not a major concern, given the scope of a workshop.
- While the authors provide some comparison to existing systems, the comparative analysis could be more comprehensive and detailed. However, the authors acknowledge the challenges in making apples-to-apples comparisons given the differences in datasets and compute resources.
- The inclusion of ablation studies or a deeper analysis of the key design choices and hyperparameters would help disentangle the impact of various components and strengthen the technical contributions.

---

### Official Review · Reviewer_c1Xk · 2024-10-08
**Efficient Reinforcement Learning for Theorem Proving: A Promising Approach with Room for Improvement**

**Rating:** 7
**Confidence:** 3

**Review:**

This paper introduces a robust baseline for reinforcement learning (RL) applied to theorem proving, achieving state-of-the-art performance while utilizing fewer training samples.

Pros:

The paper proposes a proof search algorithm that adapts the AlphaZero framework, presenting a novel approach within the domain of automated theorem proving. This innovation stands out as a key contribution.

The experimental results are compelling. The proposed algorithm significantly outperforms HTPS, requiring 13x less computational resources and establishing a new state of the art on the PutnamBench benchmark. This efficiency in both computation and performance is highly noteworthy.

Cons:

However, from the results in Table 1, the proposed method still falls short when compared to DeepSeek-Prover in terms of overall performance. It would be valuable to explore strategies for reducing the training samples required by DeepSeek-Prover, which could enhance its applicability and competitiveness in the field.

---

### Official Review · Reviewer_SUKC · 2024-10-08
**A Scalable and Efficient Reinforcement Learning Framework**

**Rating:** 6
**Confidence:** 3

**Review:**

This paper proposes a scalable and efficient reinforcement learning framework for neural theorem proving. The authors present ABEL, a system that achieves comparable performance to state-of-the-art methods while using significantly less data and computational resources. The framework use online reinforcement learning to improve data efficiency in mathematical reasoning tasks.

## Pros:
1. High data efficiency: The system achieves performance comparable to current state-of-the-art methods using only a few hundred mathematical exercises. This represents a significant advancement in sample efficiency for neural theorem proving.
2. Computational efficiency: Compared to HTPS, the proposed method reduces computational requirements by a factor of 13 while slightly improving performance. This is an impressive result that could make advanced theorem-proving systems more accessible and practical.

## Cons:
1. The paper's writing and structure need improvement. After reading the entire article, the system architecture and algorithm usage remain unclear. The absence of a clear system architecture diagram or algorithm examples, hinders full comprehension. While it's evident that the authors conducted numerous experiments, the paper would benefit from a high-level summary analyzing more about which components contribute most significantly to the impressive results.
2. The code is not open-sourced, which limits reproducibility and further exploration by the research community.

---

### Decision · Program_Chairs · 2024-10-09

Accept